# Alchemically-glazed plasmonic nanocavities using atomic layer metals: controllably synergizing catalysis and plasmonics

Shu Hu [1,2], Eric S. A. Goerlitzer [2], Qianqi Lin [2], Bart de Nijs [3], Vyacheslav M. Silkin [4,5,6] & Jeremy J. Baumberg [2] ✉

Plasmonic nanocavities offer exceptional confinement of light, making them effective for energy conversion applications. However, limitations with stability, materials, and chemical activity have impeded their practical implementation. Here we integrate ultrathin palladium (Pd) metal films from sub- to few- atomic monolayers inside plasmonic nanocavities using underpotential deposition. Despite the poor plasmonic properties of bulk Pd in the visible region, minimal loss in optical field enhancement is delivered along with Pd chemical enhancement, as confirmed by ab initio calculations. Such synergistic effects significantly enhance photocatalytic activity of the plasmonic nanocavities as well as photostability by suppressing surface atom migration. We show the atomic alchemical-glazing approach is general for a range of catalytic metals that bridge plasmonic and chemical catalysis, yielding broad applications in photocatalysis for optimal chemical transformation.

Efficient light harvesting for converting photon energies to other forms is crucial in responding to growing global energy demands[1–4]. Plasmonic nanostructures yield intense light confinement due to coupling with collective electron oscillations (surface plasmons)[5–8], which opens up promising routes for light energy conversion. Various applications have been proposed based on plasmonic nanoconstructs involving photocatalysis[2,9–11], optoelectronics[12,13] and photovoltaics[14,15]. However, control of both optical field enhancements and simultaneously their chemical activity has remained challenging.

Decay of excited surface plasmons leads to the generation of hot carriers as well as thermal effects which boost plasmonic photocatalysis[16,17]. However, the increasing temperatures lead to multiple side reactions while also reshaping the plasmonic nanostructures. Hot electronic carriers are intrinsically short-lived in plasmonic metals which limits photoconversion yields. One tactic to enhance the efficiency has been to hybridize or couple plasmonic materials with semiconductor nanostructures (termed 'antenna reactor'[18–21]) for

capturing the hot carriers. However, the enhancement is either limited by low coupling efficiency or thicker coatings of poor plasmonic materials. In addition, the surface atoms of plasmonic metals (mostly Au, Ag, and Al) have high mobility under incident light, producing rapid degradation of such nanostructures[22–24].

A successful alternative approach is to coat an ultrathin foreign metal shell around a Au core. In this borrowing strategy[25], the Au core is plasmonically active while the shell is a catalytically active metal such as Pt or Pd. This method is often used with surface-enhanced Raman spectroscopy (SERS) to track the vibrational fingerprints of species absorbed on the metal surfaces, allowing study of catalytic reactions in situ on transition metals[26–30]. An ultrathin shell (1–5 nm) is crucial for maintaining strong SERS enhancements, but is within the penetration depth for light-induced hot carriers, heat, electromagnetic (EM) fields, and surface strain, which can all accelerate reactions. Few studies can deeply consider these effects however as such composite nanostructures are very hard to reliably construct at the atomic level into

[1]Department of Physics, Xiamen University, Xiamen, China. [2]Nanophotonics Centre, Dept. of Physics, Cavendish Laboratory, University of Cambridge, Cambridge, UK. [3]Physics for Sustainable Chemistry Group, Dept. of Physics, Cavendish Laboratory, University of Cambridge, Cambridge, UK. [4]Donostia International Physics Center, San Sebastián/Donostia, Spain. [5]Departamento de Polímeros y Materiales Avanzados: Física, Química y Tecnología, Facultad de Ciencias Químicas, Universidad del País Vasco, Donostia-San Sebastián, Spain. [6]IKERBASQUE, Basque Foundation for Science, Bilbao, Spain. ✉e-mail: jjb12@cam.ac.uk

well-defined plasmonic nanocavities with consistent ultrathin metallic components that efficiently confine light.

An ideal composite material would have precision at the atomic level. In this work, we develop an alchemically-glazed nanocavity by using underpotential deposition to grow atomic monolayers of Pd onto Au facets incorporated within nanoparticle-on-mirror (NPoM) plasmonic constructs (also termed NP-on-film or MIM metasurfaces). Such well-defined nanocavities allow precise and consistent control of the metallic nanogap separation down to below 1 nm, achieving extreme coupling strengths that can mediate reactions[6,31]. Atomic layer Pd deposition (used as a first exemplar here) builds a catalytic surface with ultra-high atom efficiency which is found surprisingly to have minimal influence on plasmonic activity. We find a synergistic effect between alchemical surface and plasmonic core, inducing 30-fold increase in photocatalytic ethanol oxidation under visible light illumination. The catalytic activity of glazed NPoMs can be continuously tuned from sub-monolayer to three monolayers, saturating at 4 monolayers (4 ML). To understand these effects, we study >10,000 NPoM nanocavities, in which the ultrathin Pd monolayer actually

enhances the optical confinement of NPoMs nanocavities eliciting stronger SERS signals and red shifting the cavity modes. Ab-initio simulations confirm that monolayer Pd minimally degrades the field confinement of nanocavities. By contrast, the rate of light-induced atomic surface reconstruction (termed picocavities[7,23] and flares[32,33]) is strongly blocked by Pd monolayers, significantly increasing the photostability of such NPoMs. These findings give new insights in atomic surface chemistry of plasmonic materials and provide valuable routes to create robust and highly catalytic plasmonic architectures for photocatalysis and optoelectronics. Moreover, the generality of this glazing method makes it suited to almost all classic catalytic metals, opening a space of crucial reactions to explore.

## Results and discussion

### Monolayer Pd deposition and characterization

To uniformly glaze an atomic layer of catalytically-active Pd onto NPoMs (Fig. 1a), a template stripping method is used to fabricate atomically-smooth Au films (Supplementary Fig. 1). Citrate based wet-chemistry synthesizes typical spherical Au nanoparticles (NP) (see

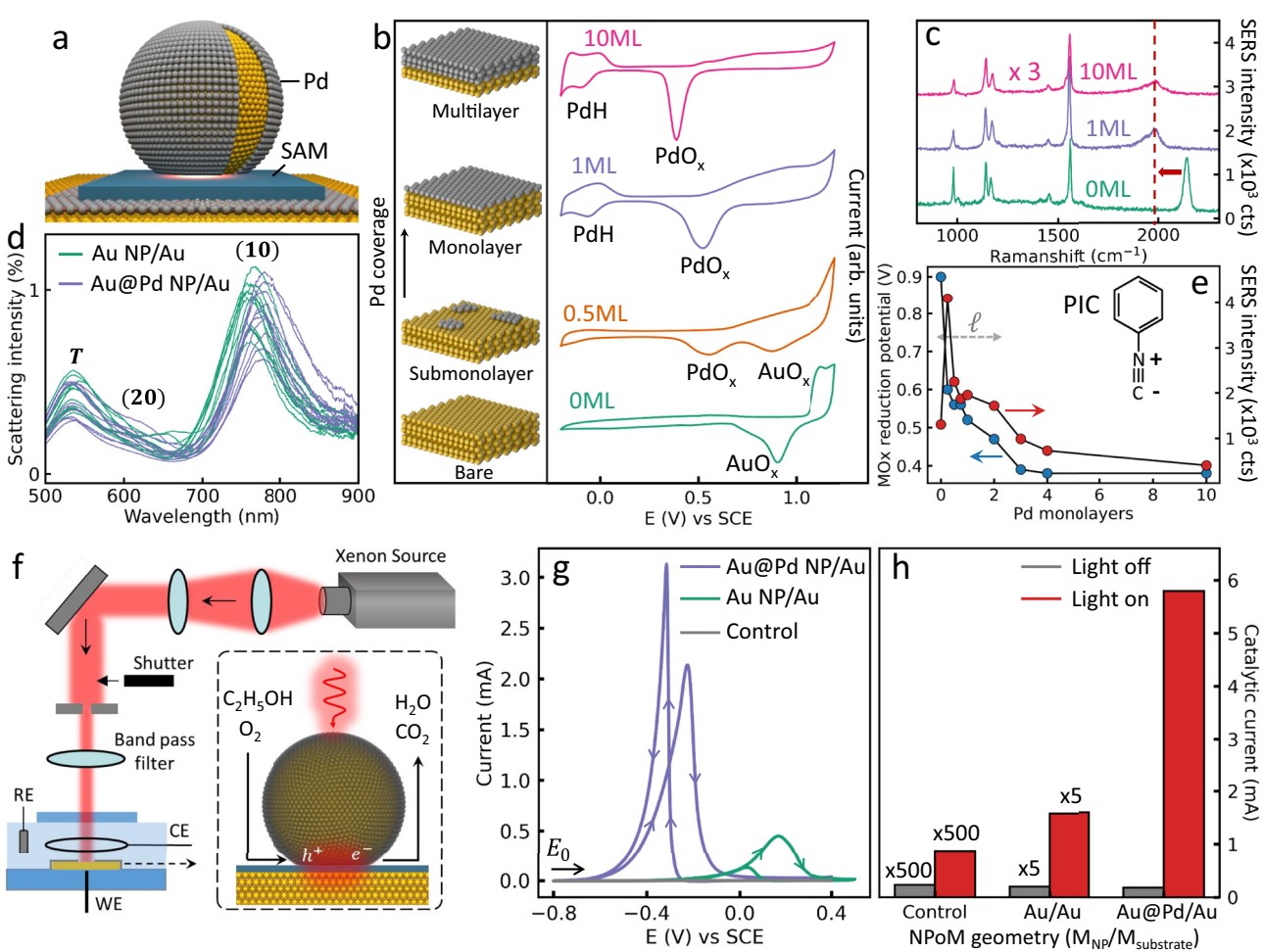

**Fig. 1 | Optical and electrochemical characterization of atomic Pd-glazed nanoparticles-on-mirror (NPoM). a** Schematic of NPoM construct integrated with a monolayer of Pd metal (grey). **b** Cyclic voltammetry (CV) measurements of Au@Pd nanoparticles with different Pd coverage. Electrolyte is 0.1 M $H_2SO_4$. CV measurements are conducted on 5 mm electrodes that encompass >$10^6$ nanoparticles. **c** SERS spectra of phenyl isocyanide (PIC) molecules adsorbed on Au@Pd nanoparticle aggregates glazed with different numbers of Pd monolayers. SERS measurements are recorded from >100 individual nanoparticles. **d** Dark field spectra of individual NPoMs before (green) and after (blue) one monolayer Pd is included. **e** Extracted $PdO_x$ reduction potential and SERS intensity of Au@Pd

nanoparticles *vs* number of Pd monolayers. Chemical penetration length $\ell$-0.6 nm. Inset shows chemical structure of PIC. **f** Setup for photocatalytic measurements of ethanol electrochemical oxidation on Pd-glazed NPoMs. **g** CV and **h** photocurrent measurements of bare Au NPoMs and monolayer Pd-glazed NPoMs using 1-dodecanethiol self-assembled monolayer (SAM) as a spacer. CV and photocurrent measured in aqueous 0.1 M NaOH and 0.1 M $C_2H_5OH$ with a scan rate and constant potential of 50 mV/s and −0.2 V, respectively. The density of NPoMs and electrode areas are the same in all measurements. Control sample is 1-dodecanethiol SAM on Au film.

Methods). Self-terminating underpotential deposition methods implemented with electrochemistry and wet-chemistry are utilized for growing monolayers of Pd on the clean Au substrate and the citrate-capped Au NPs, respectively (Supplementary Fig. 2 and Methods). To confirm monolayer Pd, we characterize films with synchrotron XPS (Supplementary Fig. 3). Since different X-ray photon energies probe different depths, comparison of peak areas from Pd $3d_{5/2}$ (shifted from the bulk position showing it is all 1 ML Pd) and Au $4d_{5/2}$ (underneath) shows the latter disappears for low penetration depths proving no bare Au surface remains (so the Pd does not reconstruct).

Cyclic voltammetry measurements show clear oxidation and reduction of Pd after the deposition (Fig. 1b and Supplementary Fig. 4). The $PdO_X$ reduction peak gradually increases as the $AuO_X$ reduction peak decreases, quantitatively tracking the Pd coverage as it is tuned from sub-monolayer to monolayer. A significant change in the reduction potential of $PdO_X$ is observed increasing from one to ten Pd monolayers (Fig. 1b) due to electronic structure hybridization between Au and Pd together with lattice mismatch induced tuning of the surface strain. These effects mainly shift the metal $d$-bands and create a surface that is more catalytically active than both bulk Au and Pd[34]. Note the $PdO_X$ reduction peak continuously shifts to lower potential with increasing number of layers, implying increasing Pd-O binding (likely because of steric and strain effects) and high quality layer-by-layer deposition. While cyclic voltammetry measurements can vary slightly between different runs, especially due to factors such as nanoparticle packing morphology, local surface interactions and stability of reference electrode, this only minimally alters the general trends observed across all these measurements.

The difference in electronic structure can also be probed by SERS (Fig. 1c) using phenyl isocyanide (PIC) molecules containing a surface-sensitive triple bond (Fig. 1e inset)[35]. This N≡C vibration significantly shifts (>150 cm⁻¹) from bulk Au to monolayer Pd (arrow), due to the difference in coordination geometries (Supplementary Fig. 3)[35]. The N≡C-Au vibration vanishes gradually with the increase of Pd coverage confirming the monolayer Pd completion (Supplementary Fig. 4). These spectral changes highlight the effectiveness of SERS in detecting subtle shifts in surface coordination and offer clear evidence of Pd deposition at the monolayer level. As such, the observed trends in the SERS spectra not only validate the presence of monolayer Pd but also provide valuable insights into the electronic and structural changes occurring at the Pd/Au interface. The SERS intensity of the phenyl ring vibration (~1560 cm⁻¹) is utilized to compare field enhancements since it is less affected by the substrate than the N≡C vibration. This SERS intensity (for 633 nm excitation) decreases as the coating increases to 10 Pd monolayers (0–10 ML, Fig. 1e), mainly because Pd is a poor plasmonic material in the visible range. However, for single monolayer Pd-glazed Au NPs an even higher SERS intensity is found than for uncoated Au NPs, as well as a small red shift in the dominant (10) coupled mode (Fig. 1d), which indicate an initial improvement rather than degradation in the optical properties[31]. To systematically reveal such anomalous effects, we tune the Pd thickness from 0–10 monolayers and measure both $PdO_X$ reduction and SERS intensities (Supplementary Fig. 4). Surface and plasmonic activities are strongly correlated, with a dramatic increase as the number of Pd monolayers decreases (Fig. 1e) giving estimated chemical 1/$e$ penetration length $\ell$~0.6 nm. For <2 ML Pd, the SERS intensity is 50%–400% higher than for bare Au, indicating a glaze of ultrathin Pd boosts both plasmonic and chemical activity of plasmonic constructs.

To quantify the photocatalytic effect of Pd-glazed NPoMs, a spectro-electrochemical setup is developed for measuring the photocurrent during chemical reactions (Fig. 1f). The white light from a Xenon lamp is collimated and filtered to wavelengths between 600–900 nm which match the NPoM resonance (Methods and Fig. 1d, f). A monolayer of 1-dodecanethiol (1-DDT) molecules is assembled (by immersion) onto the Au film as a spacer to maintain a consistent nanogap and block catalytic contributions from the bottom Au (Fig. 1g, grey = control)[36].

The key fuel cell reaction of ethanol oxidation is studied, comparing bare Au NPoMs, 1ML Pd-glazed NPoMs, and a Au film coated by the 1-DDT SAM, where the density of NPs and area of the electrode are controlled to be the same. A ~0.4 V decrease in onset potential and sixfold increase in maximum current (Fig. 1g, green and purple) show the Pd-glazed NPoMs have much higher activity for ethanol oxidation than Au NPoMs. Further photocurrent measurements are performed at constant potential to evaluate the plasmonic effects (Supplementary Fig. 5), showing 30-fold increase in photocatalytic current for Pd-glazed NPoMs compared to in the dark, compared to an eightfold increase for Au NPoMs. The Pd-glazed NPoMs are >10× more active than the Au NPoMs, and thousand-fold better than planar films. More importantly, atomic coating with Pd gives ~1800-fold increase in catalytic efficiency (normalized by the mass of Pd loading) than directly-comparable NPoM-derived antenna-reactors (Supplementary Fig. 6). These results demonstrate the synergy between catalytic surface and plasmonic core that improves the catalytic performance. Such a design not only maximizes the light utilization but also achieves extremely high atom efficiency, so reducing the cost of Pd material.

## Optical characterization of Pd-glazed NPoMs

To resolve both optical and chemical parts of this synergy, a statistical study on four types of well-defined NPoM nanocavity is performed, where one monolayer of Pd is selectively deposited on the top, the bottom or both facets of the NPoM gaps (Fig. 2a, grey). Self-assembled monolayers (SAMs) of biphenyl thiol (BPT) are utilized as the spacer to form a nanogap of 1.5 nm[37,38]. Tracking and sorting the dark-field scattering spectra of >4000 individual NPoMs (Supplementary Fig. 7) gives histograms of the spectral distribution of the dominant (10) resonant mode for each architecture. All those NPoMs at the most common (modal) wavelength are averaged to generate the characteristic spectra (Fig. 2b)[31,38].

NPoM nanocavities confine light tightly inside the metallic gap and give rise to the (10) mode (Fig. 2b). The (10) mode wavelength is thus highly sensitive to the dipole coupling strength between the facets that is set by the refractive index, gap width and facet width (that also control light confinement, optical volume roughly scales[6] as $\lambda_{10}^{-2}$). Consequently, the monolayer of Pd induces >30 nm red shifts of the (10) mode that indicate an improvement in light confinement[31,37]. This is unexpected since Pd has a poor plasmonic response (greater absorption and lower plasmonic figure of merit than Au) in the visible and higher electron density of states than Au, that should introduce blueshifts[35]. Both a contraction of the nanogap or an increasing diameter of the nanoparticle bottom facet (Fig. 2a) could induce such red-shifts. Since even 0.5 ML Pd coating onto the Au NP gives over half the 1 ML red-shift (Supplementary Fig. 8), changes in gap size are unlikely. The red shifts are also seen with other spacer molecules (Supplementary Figs. 9 and 10). The facet-sensitive higher-order (20) mode is found to closely follow the wavelength shifts of the (10) mode (Fig. 2b), which agrees with a hypothesis of increasing facet size[31]. To further confirm this, we perform similar measurements on nanodecahedron-on-mirror (NDoM) constructs which possess consistent stable triangular (111) facets[38]. These show nearly identical resonance wavelengths for each of the coupled modes (Supplementary Fig. 11) after incorporating 1 ML Pd. Together, our data imply that the increase in optical confinement for Pd-glazed NPoMs originates from Pd-induced surface reconstruction that enlarges the NP facets when they land on the SAM-coated Au film ($f_{1-4}$). Using previous parametrisation of the electromagnetic solutions in such NPoMs[39], the estimated facet sizes increase from $f_1$~5nm to $f_4$~32nm. The atomic layer Pd also induces a 30% decrease in scattering intensity and broadening in mode spectral width (Fig. 2b, c), which also relate to increasing facet size. Note

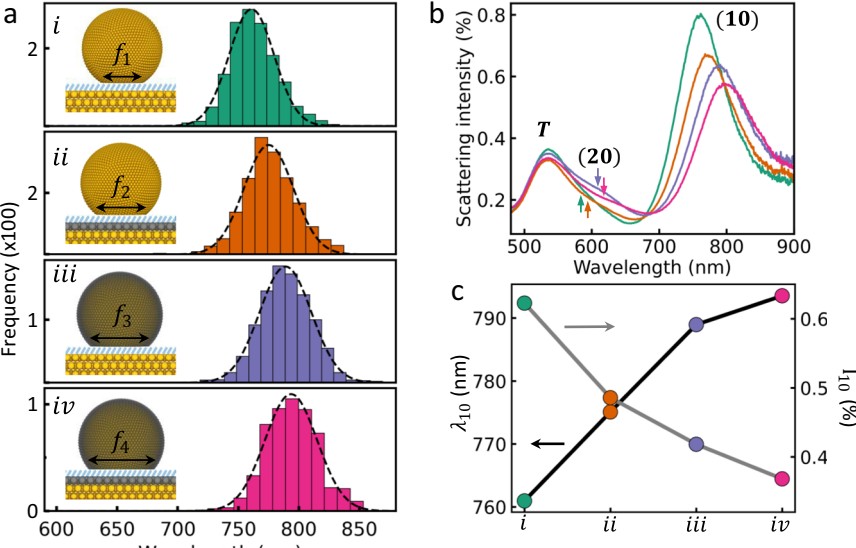

**Fig. 2 | Dark field characterization of Pd-glazed NPoMs. a** Schematic of NPoMs integrated with monolayer Pd metal (grey) on different sides of the nanogap, and the corresponding histogram of their (10) mode peak wavelengths ($\lambda_{10}$). Arrows show predicted diameter of bottom facets ($f_1 < f_2 < f_3 < f_4$), see text. **b** Comparison of average dark-field scattering spectra for the four NPoM geometries in **a**, using the central (most common) bin. Arrows mark resonance position of higher energy (20) mode. **c** Intensity ($I_{10}$) and resonance wavelength ($\lambda_{10}$) of the four different geometries in **a**, error bars below point size.

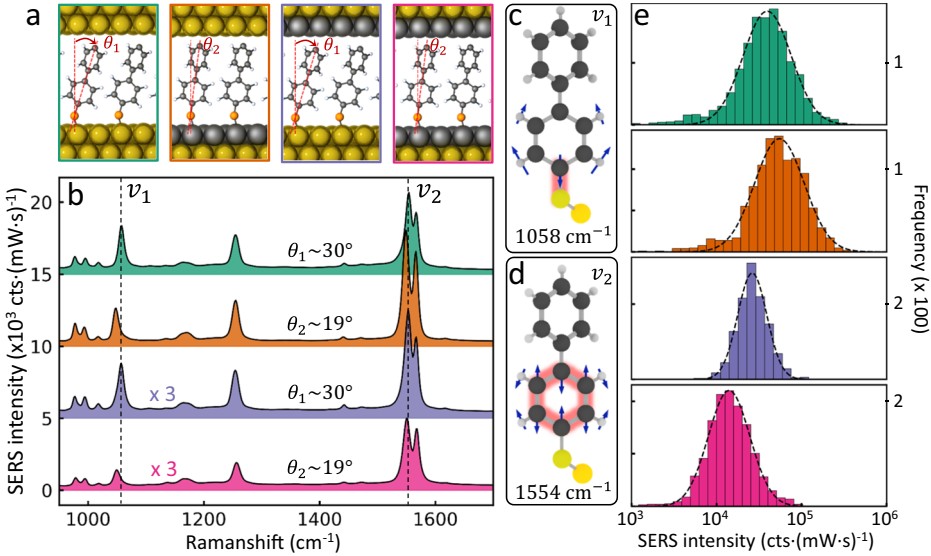

**Fig. 3 | SERS characterization of individual Pd-glazed NPoMs. a** Schematic of BPT molecules inside NPoM nanogaps with monolayer Pd glazing on different sides of the gap. **b** Average SERS spectra (each over >1000 individual constructs), with vibrations (dashed) of modes at **c** $\nu_1$ and **d** $\nu_2$ with larger amplitudes near bottom facet. **e** Histogram of $\nu_2 = 1554$ cm$^{-1}$ SERS intensities, comparing NPoM construct types from **a**.

that such information which is encoded deep within ensemble measurements, can only be resolved by detailed statistical studies on many constructs.

To explore the near field enhancements for Pd atomic glazing, SERS measurements on many thousands of NPoMs are taken for the different geometries (Fig. 3a). The spectral features are almost the same when a monolayer Pd is coated above the BPT (Fig. 3b, purple *vs* green). On the other hand, the Au-S stretch (Fig. 3c; $\nu_1 = 1058$ cm$^{-1}$) shows an obvious vibrational red shift of 10 cm$^{-1}$ (Supplementary Fig. 12) with monolayer Pd on the bottom substrate (Fig. 3b, orange and pink *vs* green). Comparing with density functional theory (DFT) simulations (Supplementary Fig. 13) agrees that there is greater charge transfer between Pd and BPT molecules than with Au, shifting the vibrational energy. Similar changes can also be observed for the

bottom ring vibration (Fig. 3d; $\nu_2 = 1554$ cm$^{-1}$) when close to the Pd atoms, with the top ring mode much less affected. This modified binding between BPT and Pd is found to slightly decrease the molecular tilt angle in DFT simulations from ~30° to ~20° (Supplementary Fig. 13e, f).

NPoMs which are Pd-glazed on the bottom facet (Fig. 3a, orange) show ~50% higher SERS signal for all lines (Fig. 3b, e). This is unlikely to arise from gap or facet size changes since the enhancement does not depend on the detuning of resonance wavelength of the coupled mode from the laser (Supplementary Fig. 14). The high surface coverage of thiol SAMs on both Au and Pd makes improbable a 50% increase in number of molecules probed (with the same enhancement seen also for different molecules, Supplementary Fig. 15). The tilt angle reduction would geometrically account only for 15% increase in packing.

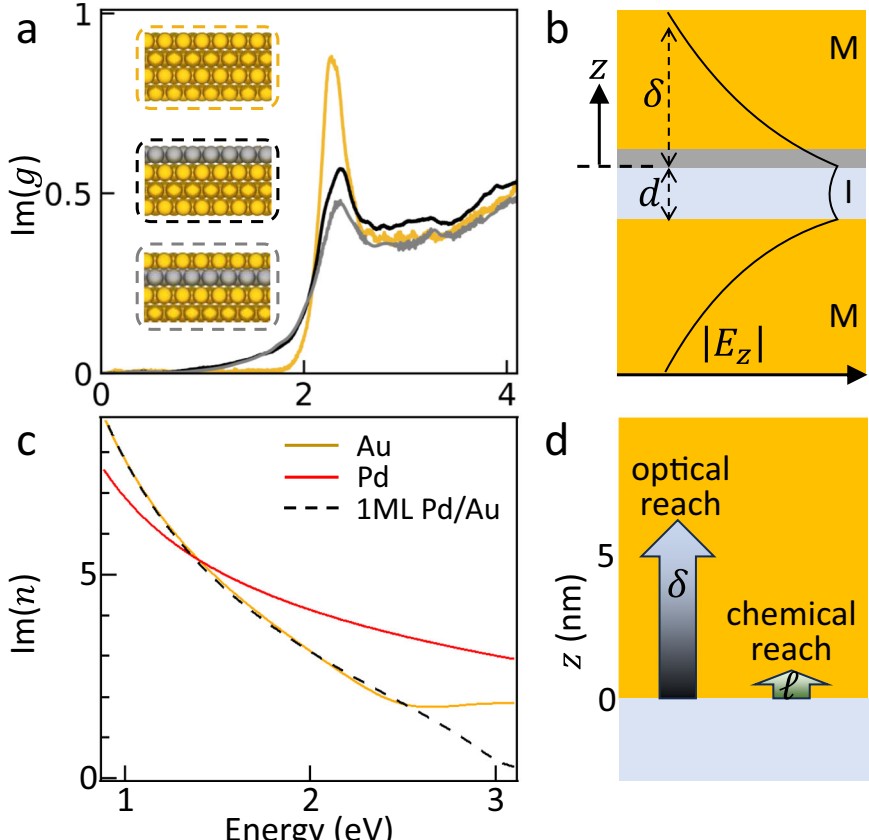

**Fig. 4 | Metal-insulator-metal (MIM) simulations. a** Loss function from ab-initio TDDFT of bare Au, 1 ML Pd on Au, and 1 ML Pd underneath the top Au atomic monolayer. **b** Magnitude of vertically-polarised optical field $|E_z|$ in a 1.1 nm ($d$) nanogap for bare Au on either facet, showing field penetration into the metal ($z$). **c** Imaginary part of the refractive index Im{$n$} for bulk Au, Pd, and for weighted average of 1 ML Pd coated facet (using the gap plasmon mode profile in **b**). **d** Schematic depth-dependence for optical field penetration ($\delta$) and chemical activity ($\ell$) *vs* depth inside the metal (see text).

Instead we suggest this SERS increase comes from charge-transfer induced enhanced polarizabilities that increase the Raman cross section (a type of non-resonant chemical enhancement[40]). By contrast, Pd-glazing the upper facet decreases the SERS intensity by 1.5-fold (Fig. 3e). A tentative explanation is that the tilt of the BPT molecules in this case increases from 30° to 38° from the surface normal so that the Raman dipole becomes misaligned with the $z$-polarised optical fields (perpendicular to the facets). Nevertheless, we note that the SERS intensities still exceed $10^4$cts·(mW·s)$^{-1}$, enabling precision real-time vibrational spectroscopy combined with exquisitely-functionalised catalytic activity. We also note that the SERS intensity distributions across different individual NPoMs vary less when Pd-glazed on the top nanoparticle (Fig. 3e), suggesting this is a source of inconsistency in Au NPoMs that can now be overcome.

### Plasmonics of Pd-glazed Au

Theoretical models of Pd monolayers on Au confirm the intuition developed above. We first use ab-initio time-dependent density functional theory (TDDFT) to quantify the effect on surface plasmons of Pd monolayer coatings on Au. Comparing the loss functions[41] shows that the surface plasmon at ~2.3 eV weakens by only 30% when a Pd monolayer coats the Au (Fig. 4a). By contrast, for bulk Pd surfaces, the system is lossy at all photon energies and there is no plasmon resonance. The weakening of the surface plasmon is slightly stronger if this Pd monolayer is placed underneath the top Au atomic layer (grey), showing the ability to further atomically engineer surface properties. As a comparison, a classical multilayer approach using the Au,Pd bulk permittivities agrees with the equivalent volume-weighted permittivity $\bar{\varepsilon}_m$ combining bulk Au and Pd in the ratio of their thicknesses within the

skin depth, to define the complex refractive index that sets the confined gap plasmon (Fig. 4b,c). Although bulk permittivities show the surface plasmon on bare Au at ~2.3 eV is half as lossy as on Pd, in the nanogap the coupled plasmon resonance is redshifted below 1.5 eV (Fig. 1d), where Pd actually outperforms Au (Fig. 4c). For a $d = 1.1$ nm nanogap between Au facets with gap index $n_g$, the metal-insulator-metal (MIM) plasmon waveguide mode penetrates $\delta = dn_g^2/2\bar{\varepsilon}_m \sim 5$ nm into the Au[6], which shows why the effect of only an atomic monolayer of Pd is indeed small (Fig. 4c dashed). Comparing penetration $\delta$ into the metal (Fig. 4d), for which the optical field sees predominantly Au, with the surface chemistry reach $\ell$ which sees predominantly surface Pd, shows how independent optimisation of these properties is feasible.

### Photostability and light induced surface dynamics

A further crucial aspect for photochemistry is the photostability of the facet when glazed by an atomic layer of foreign metal. Time-dependent SERS measurements are thus performed on these constructs. Transient events from picocavities[7,23] (adatoms) and flares[32,33] (adlayers) originate from the light-induced reversible surface dynamics of Au, as frequently seen for the bare Au NPoMs (Fig. 5a). Picocavities are atomic protrusion inside the NPoM metallic gap formed by pulling out a single atom from the surface, and give extreme light confinement that generates a huge increase in SERS signal. The generation rate of picocavities thus depends on the energy barrier to pull out a single atom. Irradiation with μW light is sufficient to stochastically create many picocavities on Au NPoMs due to the high mobility of Au atoms at room temperature. However, these are almost completely suppressed by glazing a monolayer Pd on the bottom nanogap facet (Fig. 5b). This

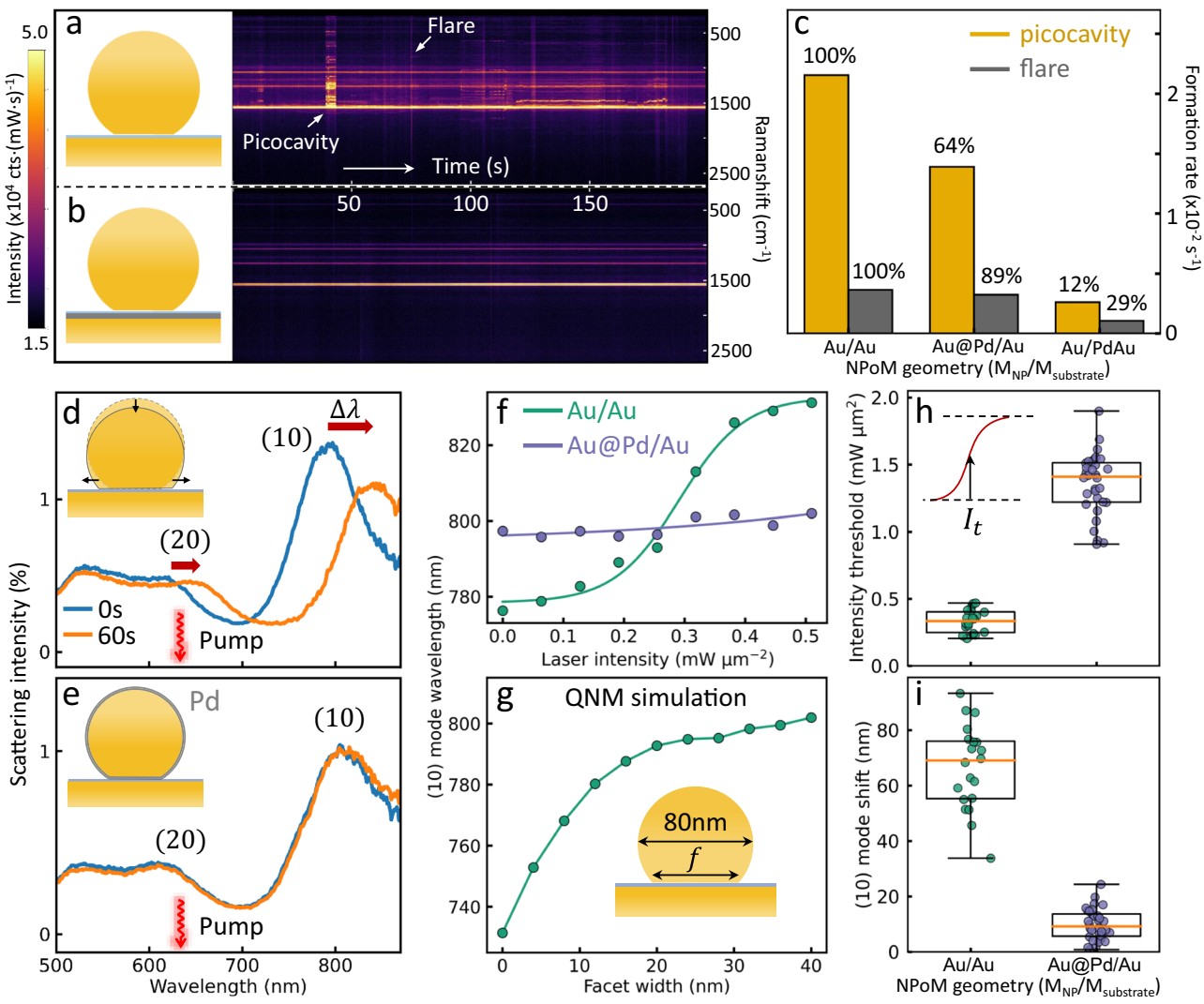

**Fig. 5 | Surface dynamics and photostability of Pd-glazed NPoMs. a**, **b** Time-dependent SERS spectra of **a** bare Au NPoMs and **b** with monolayer Pd on Au substrate, intensity scale bar on left, picocavities and flares indicated. **c** Formation rate of picocavities and flares of Pd-glazed NPoMs for 633 nm laser irradiation (130 μW/μm²), % shows rates for observing picocavities and flares normalized to Au/Au. **d**, **e** Scattering spectra of NPoMs before (blue) and after (orange) 633 nm laser illumination (200 μW/μm²) for 60 s (**d**) without and (**e**) with protection of Pd monolayer. Δλ shows resonance energy shift. **f** Wavelength shift of $\lambda_{10}$ modes *vs* laser intensity, lines are sigmoidal fits. **g** Quasinormal mode simulated wavelength shift of $\lambda_{10}$ mode *vs* facet width $f$ of NPoM. **h** Extracted laser intensity threshold ($I_t$) and **i** laser-induced total $\lambda_{10}$ spectral shift for >20 NPoMs with (purple) and without (green) monolayer Pd. Laser intensity dependent measurements performed separately on each NPoM. Error bars show distribution of intensity thresholds and spectral shifts among different NPoMs.

shows that the atomic layer of Pd not only develops a catalytically-active surface but also blocks extraction and movement of the more mobile Au atoms by increasing the binding energy and energy barrier for surface hopping. To quantitatively characterise this blocking effect, the formation rate of picocavities and flares is extracted from a million SERS spectra using a filtering algorithm[23] (Fig. 5c). This shows flares are more difficult to form (∼ sevenfold lower rate) than picocavities, as expected for a flare formation mechanism that requires lifting a layer of many atoms[42]. Distinct changes in formation rate of both flares and picocavities are observed with the Pd glazing either on top or bottom of the nanogap (and where the sum of these rates matches the bare Au NPoMs). This suggests that picocavities (>60% from bottom) and flares (>80% from bottom) on different surfaces can be selectively blocked by depositing a monolayer Pd.

Light-induced migration of facet atoms is frequently observed as a slow red-shift of the plasmonic gap modes (Fig. 5d). This nanoparticle facet instability[43,44] limits many practical plasmonic applications. The energy shift however disappears in Pd-glazed NPoMs (Fig. 5e) across a

broad intensity range (Fig. 5f). The intensity-dependent resonance wavelength shifts allow extraction of an illumination threshold and saturated spectral shift of the (10) mode (Fig. 5h, i). The Pd-glazed NPoMs show >400% higher threshold (above our power range) and 10 times less spectral shift, allowing much higher laser powers while maintaining the spectral position of the nanocavity modes. The photostability of plasmonic nanocavities is thus significantly improved with a monolayer of Pd, without compromising the plasmonic properties. This is of more general utility.

Using facile underpotential deposition allows for glazing an atomically-thin foreign metal onto the facets of plasmonic nanocavities. By performing statistical studies on thousands of individual NPoM structures, we show an atomically-thin foreign metal glazed inside plasmonic NPoM nanogaps improves both catalytic and optical performance, as well as their photostability. We observe synergistic effects of catalytically active Pd and plasmonically active Au which bolsters both the photocatalytic activity and optical confinement of the nanocavities. We show the chemical enhancement of the

**Table 1 | Alchemical comparison of monolayer catalytic metals on bulk Au substrate**

| Metal | Lattice mismatch | Atomic radius [Å] | Electro-negativity (Pauling) | Experimental observation[a] | Solvent | Plasmonic properties in visible | Typical applications |
|---|---|---|---|---|---|---|---|
| Au | 0.00% | 1.74 | 2.54 | – | – | Good | CO oxidation |
| Ag | 0.17% | 1.65 | 1.93 | Yes | Aqueous | Good | Propylene Epoxidation |
| Pt | 3.92% | 1.77 | 2.28 | Yes | Aqueous | Bad | Methanol oxidation |
| Pd | 4.82% | 1.69 | 2.20 | Yes | Aqueous | Bad | Ethanol oxidation |
| Rh | 7.23% | 1.73 | 2.28 | Yes | Aqueous | Bad | Hydrogenation reactions |
| Ir | 6.23% | 1.80 | 2.20 | Yes | Aqueous | Bad | C-H activation |
| Cu | 12.81% | 1.45 | 1.90 | Yes | Aqueous | Good | $CO_2$ reduction |
| Ni | 15.73% | 1.49 | 1.91 | Yes | Aqueous | Bad | Cross-Coupling Reactions |
| Fe | 42.27% | 1.56 | 1.83 | Yes | Aqueous | Bad | Haber-Bosch Process |
| Ru | 50.72% | 1.78 | 2.20 | Yes | Aqueous | Bad | Ammonia synthesis |
| Co | 62.67% | 1.52 | 1.88 | Yes | Aqueous | Bad | Fischer-Tropsch synthesis |
| Zn | 53.03% | 1.42 | 1.65 | Yes | Aqueous | Bad | Click Chemistry |
| Re | 47.71% | 1.88 | 1.90 | Yes | Aqueous | Bad | Alkene Epoxidation |
| Ti | 38.21% | 1.76 | 1.54 | Yes | Ionic liquid | Bad | Hydroamination |

[a]for more details, see Supplementary Table 1.

monolayer Pd compensates minimal negative impacts on plasmonic properties that overall enhances the optical response of the system. Furthermore, the atomic layer glazed Pd efficiently blocks the formation of picocavities and flares by inhibiting migration of surface atoms, which creates an alternative approach for controlling surface chemistry at the atomic level.

A promising direction for future research is the exploration of catalytic activity at the single-molecule level within NPoMs. Tracking single-molecule catalysis would provide invaluable insights into the detailed mechanisms driving plasmon-enhanced catalysis, enabling more precise control and optimization of catalytic reactions. One potential approach to achieve this is by controlling picocavities generated inside the NPoM geometry. The mechanisms governing picocavity formation remain complex, making this a challenging yet highly intriguing avenue for future research. Despite such challenges, our current findings, which highlight the amplified catalytic and optical activity of glazed NPoMs, provide a strong foundation for advancing to single-molecule studies. As the technology and understanding of picocavity formation improves, we foresee exciting possibilities for further resolving plasmonic catalysis at the single-molecule scale. We also highlight the need for considering atom efficiency in discussions of enhanced catalytic activities, which is often difficult to achieve in less controlled atomic morphologies.

Although we focus here on Pd glazing as an exemplar, this approach is general. An alchemical comparison (Table 1) shows almost all popular catalytic metals are accessible, which can thus access wide admixtures of chemical and plasmonic properties in catalysing crucial chemical reactions. So far, we already successfully produced Pt, Ag and Cu analogous 1 ML systems (for SERS characterisation, see Supplementary Fig. 16). Exploring additional catalytic reactions, the Pt- and Cu- 1 ML systems show 26- and 37-fold increases in photocatalytic current for hydrogen evolution reactions and $CO_2$ reduction, respectively (Supplementary Fig. 17). Underpotential deposition is primarily driven by the electronegativity difference between metals. The higher electronegativity metal is typically used as a substrate and the lower electronegativity metal wets the surface and forms two-dimensional layers[45,46]. A large electronegativity difference ($\mathcal{X}$) creates strong binding energy $\mathcal{D} \propto (\mathcal{X}_{\text{substrate}} - \mathcal{X}_{\text{surface}})^2$ between substrate and surface metal[47], which is essential for depositing foreign metal atoms at more positive potential (underpotential) than the Nernst potential

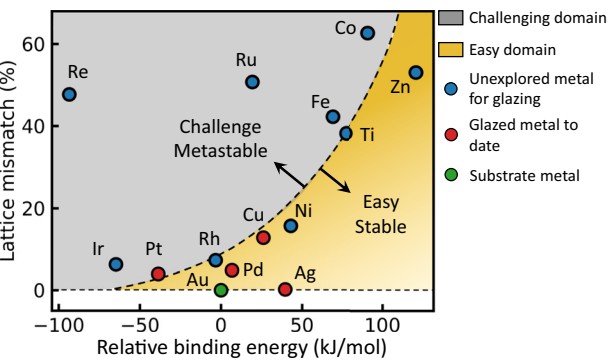

**Fig. 6 | Alchemical space for atomic-glazing on Au substrates.** Relative binding energy is energy difference ($\Delta\mathcal{D}$) between glazed metal M-Au ($\mathcal{D}_{M-Au}$) and M-M ($\mathcal{D}_{M-M}$) bond. Large $\Delta\mathcal{D}$ ensures underpotential deposition occurs at more positive potential with good separation from overpotential deposition. Yellow shaded region shows metals that most easily form monolayer glazed films of high quality. Grey region shows challenging domain, where surface modification is essential to alter deposition potential and stability of atomic glazed films. Red points indicate 1 ML glazed NPoMs to date.

(overpotential). Hindering atomic layer glazing is the lattice mismatch which induces surface tension of the monolayer film, and determines the stability and chemical activity of the film, although a wide alchemical space is accessible (Fig. 6). Surface tension can be released using metals with smaller radius atoms—for instance, it has been shown that Cu can deposit uniform monolayers on Au even with its larger surface lattice mismatch[30,48]. Because of its large electronegativity and radius, as well as its chemical stability, Au makes an excellent substrate for glazing many metals (Table 1). While changes in $d$-band position for these glazed monolayers can shift chemical affinities, for example permanently binding products and thus leading to poisoning, this is not so far evidenced from the improved reactivities achieved.

Our studies thus open up a wide variety of applications across plasmonics and photochemistry. We envision construction of robust and highly photocatalytically active plasmonic nanostructures with

ultrahigh atom utilization efficiency for photocatalysis, nanophotonics and optoelectronics.

## Methods

### Chemicals

Gold (III) chloride trihydrate (HAuCl$_4$·3H$_2$O, ≥99.9%), palladium(II) chloride (PdCl$_2$, 99%), sulfuric acid(H$_2$SO$_4$, 95.0–98.0%), citric acid (99.5%), ascorbic acid (AA, ≥99%), sodium citrate tribasic dihydrate (≥99%), biphenyl-4-thiol (BPT, 97%), and ethanol (99.5%) were purchased from Aldrich-Merck. All chemicals are used without further purification. Milli-Q ultrapure water (resistivity 18.2 MΩ·cm at 25 °C) is used in all experiments. All glassware and stir bars are cleaned with aqua regia, piranha solution and rinsed with ultrapure water prior to use.

### Synthesis of Au@Pd nanospheres

Gold nanoparticles (NPs) are synthesized using a multi-step approach to enable better morphology control, according to published procedures[49]. 2.4 mL 1% HAuCl$_4$ is diluted in 100 mL ultrapure water and boiled in a round bottom flask while stirring with condensate reflux. 2 ml 1% sodium citrate is then quickly inject into the flask. Reacting for 40 mins yields a brick red color solution of 45 nm (diameter) Au NPs as seeds. 7.5 ml Au NPs solution is diluted with 50 ml ultrapure water in a round bottom flask when it is cooled down to room temperature. 380 μl 1% ascorbic acid (AA), 75 μL 1% sodium citrate and 220 μl 1% HAuCl$_4$ are slowly injected (one drop per ten seconds) into the diluted seed solution in an ice bath while stirring, which yields the Au NPs of 80 nm mean diameter.

Au@Pd NPs are prepared by adapting established approaches from the literature[29]. 10 mL as-synthesized 80 nm Au NPs and 12 μL 1% AA are added in a round bottom flask and kept in an ice bath for 30 min. 66.7 μL 1 mM H$_2$PdCl$_4$ is slowly injected into the flask while stirring. Leaving the reaction overnight yields a monolayer Pd shell on the Au NPs. The thickness of the shell can be tuned by altering the volume of H$_2$PdCl$_4$ solution. The growth mechanism is similar to electrochemical underpotential deposition with the concentration of AA controlling the reduction potential (Supplementary Fig. 2). The epitaxial growth of Pd on Au NPs to make uniform monolayers has been demonstrated and characterized in our studies using electrochemistry (Supplementary Fig. 4), as well as TEM measurements in the literature[29,50].

### Monolayer Pd electrodeposition and characterization

Monolayer Pd is deposited on ultra-smooth Au films (Supplementary Fig. 1) using electrochemical underpotential deposition[51]. The Au film is immersed in the electrolyte of 0.1 M H$_2$SO$_4$ and 1 mM H$_2$PdCl$_4$ under potential control. A linear voltage sweep is applied to determine the potentials for underpotential and overpotential deposition (Supplementary Fig. 2). The coverage and layers of Pd on Au film are tuned by controlling the deposition charge while holding the potential at the underpotential deposition peak. Details of the cell are shown in Supplementary Fig. 18.

The exceptionally narrow 1–2 nm nanogap between the 80 nm Au nanoparticle and the bulk substrate precludes the use of high-resolution transmission electron microscopy (HR-TEM) for direct structural analysis. The geometry of this system makes it extremely challenging to resolve monolayer atoms within the nanogap at atomic resolution. Additionally, the high-energy electron beam required for optimal resolution in HR-TEM can cause irreversible damage to the delicate monolayer structures, raising concerns about the reliability of the data obtained. Furthermore, HR-TEM provides only sectional views of atomic phases, limiting our ability to capture the overall lateral surface characteristics, especially considering the statistical variations across a large number of NPoM geometries. Techniques like low-dose

imaging and focused ion beam milling, while useful in some contexts, are not well-suited for our specific system due to the complexity and scale of the nanogap.

Instead, we employ cyclic voltammetry and synchrotron XPS, which are well-suited for bulk-sensitive characterization. Cyclic voltammetry tracks Pd coverage from sub-monolayer to multilayer regimes, while XPS confirms the chemical state and uniformity of the deposited monolayers. These techniques, combined with SERS measurements and statistical analyses across multiple samples, provide a robust and reproducible approach to characterizing the system.

### Electrochemical photocatalytic measurements

A setup (Fig. 1f) is built for measuring the photocurrent during electrochemical reactions. A Bentham DTMc300 Xenon white light is collimated by two lenses, passes through a shutter, slit, and band pass filter (600–900 nm) before illuminating the electrode. All the measurements use the same power of 0.3 W·cm⁻². The surface area of the working electrode (Au film) is controlled to be diameter 5 mm by an O ring. The reference is a saturated calomel electrode and the counter electrode is Pt wire. The electrolyte is 0.1 M NaOH and 0.1 M ethanol.

### NPoM sample preparation

Synthesized nanospheres of 80nm diameter of Au and Au@Pd are centrifuged twice and cleaned with ultrapure water to remove surfactants as much as possible. BPT, BPDT, PDI, and 1-DDT molecular layers are assembled by immersing Au films into 1 mM solutions of these molecules in ethanol for 16 h. Then, the substrates are rinsed with copious amount of ethanol and dried with nitrogen gas. The cleaned NPs are drop-cast on top of BPT and 1-DDT SAMs for 10 s and 60 s to create NPoMs, rinsed with ultrapure water and dried with nitrogen afterwards. Au and Au@Pd nanospheres that are used for comparison are from the same batch. In photocatalytic measurements, the NPoM density is optimized to maximize catalytic performance while minimizing aggregation (Supplementary Fig. 19). For optical measurements, a lower density ensures reliable acquisition of single particle spectra.

### Synchrotron X-ray photoelectron spectroscopy measurements

Depth-dependent X-ray photoelectron spectroscopy (XPS) is performed at the Diamond Light Source synchrotron facility (Oxfordshire, UK) on beamline I09 with soft X-ray photon energies of 200, 450, 750, and 1100 eV. These correspond to photoelectron kinetic energies with inelastic mean free paths (IMFPs)/escape depths of 0.39, 0.63, 0.88, and 1.24 nm assuming gold as the dominating element. Spectra were analysed using CasaXPS (www.casaxps.com), and the binding energies (BE) were calibrated using the Au 4f$_{7/2}$ peak at 83.96 eV and the Au 4d$_{5/2}$ at 335 eV. Analysis of the survey XPS spectrum reveals minimal traces of carbon and potential oxygen, with no additional elements visible.

### Single nanoparticle scattering and SERS measurements

The SERS and scattering spectra measurements are performed in a home-built confocal Raman microscope, as discussed in detail in former studies[38]. The setup allows automatic single particle identification that provides statistical spectroscopy data from a large number of particles. The integration time for scattering and SERS spectra are 1 s. All the SERS measurements are performed with a 633 nm laser.

### Ab initio simulations

The calculations are carried out applying linear response theory, with full inclusion of the electron band structure by means of a first-principles pseudopotential approach in a supercell scheme. In the response calculations, realized with an in-house code, the adiabatic local density approximation (ALDA) is employed to account for the exchange-correlations.

## Density functional theory calculation

Molecules are modeled with covalent-bonded single Au or Pd atom to calculate Raman spectra. Gas-phase geometry optimizations are performed with no symmetry restrictions. B3LYP hybrid functional and def2SVP basis set are used with Grimme's D3 dispersion correction with Becke-Johnson damping. The UltraFine integration grid is used to increase accuracy of calculations. Calculations are carried out with the Gaussian09 program package. All calculated spectra are scaled by a factor of 0.97.

## Data availability

All data in the current study are available from the authors upon request.

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

## Acknowledgements

We thank valuable input from Javier Aizpurua and Ruben Esteban. We thank the Diamond Light Source for access to beamline I09 (under proposal SI34784-1) and I09 beamline team Pardeep Kumar Thakur and Tien-Lin Lee. Further, we thank Reshma R Rao and Mary P Ryan for help in measurement, as well as Elle Wyatt, Rakesh Arul, Ishaan Lohia, Sarah Sibug-Torres, Marika Niihori, for beamline support. We acknowledge funding from EPSRC (EP/L027151/1, EP/X037770/1, EP/X023443/1 and EP/Y008162/1), and ERC (Project No. 883703 PICOFORCE and 861950 POSEIDON). S.H. acknowledges funding from the Fundamental Research Funds for the Central Universities (Xiamen University: No. 20720240137). E.S.A.G acknowledges support from the German National Academy of Sciences Leopoldina (LPDS 2022-01). B.d.N acknowledges support from a Royal Society University Research Fellowship URF \R1\211162, and EPSRC (EP/Y008294/1). V.M.S. acknowledges financial support by Grant PID2022-139230NB-I00 funded by MCIN/AEI/10.13039/501100011033.

## Author contributions

S.H. and J.J.B. conceived and designed the experiments, and analysed the data. S.H. performed the experiments with E.S.A.G. helping with electrochemical deposition. E.S.A.G. conducted the synchrotron X-ray spectroscopy measurements. J.J.B implemented the MIM simulations, while V.M.S. carried out the TDDFT calculations. Q.L. carried out the DFT simulations and supported the electroanalysis. B.d.N. supported the automatic SERS measurements. The manuscript was written with contributions from all authors.

## Competing interests

The authors declare no competing interests.
