## [Transparent Peer Review file · Nature Communications]

Alchemically-glazed plasmonic nanocavities using atomic layer metals: controllably synergizing catalysis and plasmonics

Corresponding Author: Professor Jeremy Baumberg

Version 0:

Reviewer comments:

Reviewer #4

(Remarks to the Author)

Shu, H., et al. investigated a NPoM system utilizing an alchemically-glazed plasmonic nanocavity for the ethanol oxidation reaction. After a careful review of the paper, I share concerns regarding its novelty, significance, practicality, and the robustness of the structural evidence presented. While the study provides incremental findings on the application of the NPoM system to ethanol oxidation as a model reaction, the overall quality and broader impact of the work appear to fall short compared to other studies on similar topics published in this journal. Therefore, I recommend rejecting the manuscript for publication in Nature Communications.

- Similar ideas in previous study of the antenna-reactors, and core-shell structure have been explored, with precise control over the metallic dimensions, including shell thickness (Linic et al.), gap size (Halas et al.; Cortes et al.; Jaw-Min Nam et al.), large area (Cortes metasurface and supercrystals), and coupling strength (Misawa et al.), achieved through techniques such as lithography and colloidal self-assembly. Among these studies, the correlation between optical enhancements, reaction efficiency, and surface chemistry has been better understood. The authors claim there may be room that the optical enhancement be higher than previous study (which I highly doubt in a rigorous manner of fair comparison), but at least the impact on the kinetic characterization of the catalytic performance is lacking in this article as the support.
- The electrochemical cyclic voltammetry (CV) and synchrotron X-ray photoelectron spectroscopy (XPS) are valuable bulk analysis techniques but do not provide spatial resolution down to sub-atomic layers. In contrast, high-resolution transmission electron microscopy (HR-TEM) offers a more direct and feasible approach for achieving the high spatial resolution required, providing crucial evidence to support the claim. Although HR-TEM imaging can be challenging, it remains a reasonable and effective technique for this purpose. Additionally, low-dose imaging and focused ion beam (FIB) milling are well-established techniques that can significantly aid in imaging such metallic structures.
- Table 1 is more of a general experience rather than a well-designed guide for the application. Engineered atomic transition metal layers, depending on their local d-band structure, can effectively shift the chemical affinity for different reactions. This can cause the surface to switch from being reactive to behaving as a poison, as described by the Sabatier principle.

Reviewer #5

(Remarks to the Author)

The authors present a highly stable plasmonic nanocavity (NPoM) system that, distinct from prior picocavities, employs atomic layer deposition of Pd via an electrochemical approach. This system demonstrates stable near-field enhancements combined with the catalytic activity of Pd, achieving a remarkable 30-fold increase in photocatalytic ethanol oxidation under visible light illumination.

This review consolidates two previous reviews along with the authors' rebuttal. Below, I address whether the authors have satisfactorily responded to the previous referees' comments and provide additional insights.

Overall, the work represents a significant contribution to the field of plasmonic nanocavity systems, demonstrating innovative methods and promising catalytic applications. With minor revisions and additional clarity on the points raised below, the manuscript would be even more robust and impactful.

comments

Have the authors satisfactorily addressed the previous referees' comments?

The authors have made commendable efforts to address key critiques raised by Referee 1 and Referee 2, particularly concerning structural characterization limitations, the novelty of the NPoM system, and the justification of catalytic performance enhancements. However, some critiques, especially those by Referee 2 regarding the indirect nature of characterization methods and structural rigor—remain only partially resolved. Further experimental validation or additional discussion would strengthen the manuscript.

Comments and Suggestions

1. Cyclic Voltammetry and Monolayer Calibration

I agree with the referees that connecting cyclic voltammetry data to discrete, stepwise nanostructural changes from sub-monolayers to multilayers remains challenging. Without error bars, it is difficult to ascertain whether these surfaces are uniform or exhibit variations at the level of individual NPoMs. Additionally, the relationship between monolayer coverage and the observed signals may vary between experiments, raising questions about how this approach can be calibrated into a reliable "monolayer gauge." While the authors make a strong case for monolayer control using self-terminating underpotential deposition (supported by relevant literature), a more detailed discussion of this methodology's limitations and its reproducibility would be beneficial. The SERS spectra effectively confirm the presence of Pd (mono)layers, but the data could be further contextualized to enhance clarity.

2. Clarity on Experimental Scale and Scope

It would be helpful if the authors clarified how many NPoMs are involved in each of the different measurements (electrochemistry, SERS, XPS) - sometimes I found this to be not very clear.

3. NPoM Density and Catalytic Activity

Does the density of NPoMs influence photocatalytic activity? How was the working density of NPoMs controlled in these experiments? Has the spatial distribution of particles been characterized to ensure uniformity and to rule out potential aggregation effects?

4. Structural Characterization and Facet Size

The authors infer significant increases in facet size from indirect measurements, which is an important observation. However, direct imaging of these nanoparticle formations—possibly after dislodging them from the substrate—could provide more definitive evidence. Have the authors considered pursuing this approach, or is it technically prohibitive?

5. Single-Molecule Turnover and Catalysis

Their SERS data suggests significant potential for monitoring single NPoM systems. In the future, the authors could explore the possibility of tracking catalytic activity at the level of single NPoMs, perhaps even down to single-molecule turnover. This could require integrating controlled picocavities into the current system. A brief discussion of this direction could enrich the manuscript's outlook.

Conclusion

The authors have made substantial progress in addressing the referees' concerns and have provided thoughtful justifications and additional data in most areas. However, some points, particularly those regarding structural characterization, the reproducibility of monolayer calibration, and the influence of NPoM density on catalytic activity, warrant further experimental exploration or discussion. Overall, the work represents a significant contribution to the field of plasmonic nanocavity systems, demonstrating innovative methods and promising catalytic applications. With minor revisions and additional clarity on the above points, the manuscript would be even more robust and impactful.

Reviewer #6

(Remarks to the Author)

I have carefully reviewed the revised manuscript and acknowledge the authors' efforts to address the comments raised during the review process. They have responded thoroughly to the majority of the feedback, and their revisions have significantly improved the overall quality of the manuscript. While some of the issues remain unresolved, these do not substantially detract from the core contributions and clarity of the work.

In my opinion, the manuscript is now suitable for publication. The authors have demonstrated a commitment to enhancing their study, and the remaining concerns are either minor or beyond the scope of the current revision. I recommend acceptance at this stage.

Version 1:

Reviewer comments:

Reviewer #4

(Remarks to the Author)

After a careful and thorough review, including cross-review of the updated draft, I find that the authors have effectively addressed most of my concerns and questions. I appreciate their efforts and contributions to the field, as well as their dedication to clarifying and resolving the concerns raised by all six reviewers. Overall, the authors present a unique and insightful platform for plasmonic catalysis from an atomistic design perspective. Therefore, I recommend publication in Nature Communications in its current form, without the need for further revision.

Reviewer #5

(Remarks to the Author)

I would like to thank the authors for addressing all of my comments. The manuscript is ready for publication.

Response to reviewers:

We are delighted that reviewers #5 and #6 are happy to see this published in Nature Communications, . Reviewer #6 recognizes we *'responded thoroughly to the majority of the feedback, and their revisions have significantly improved the overall quality of the manuscript'*. They note that *'while some of the issues remain unresolved, these do not substantially detract from the core contributions and clarity of the work'*. We also appreciate reviewer #5's acknowledgement we *'satisfactorily responded'* with *'additional insights....significant contribution to the field'* and *'innovative methods and promising catalytic applications'*. We understand reviewer #4's concerns regarding the novelty of our approach and the need for stronger structural evidence, but we also note that reviewer #5 emphasises we *'made commendable efforts to address key critiques raised by Referees 1 and 2, particularly concerning structural characterization limitations, the novelty of the NPoM system, and the justification of catalytic performance enhancements'*. We have further refined the manuscript to clarify these issues and to highlight the broader implications for the field.

Reviewer 4

1. Similar ideas in previous study of the antenna-reactors, and core-shell structure have been explored, with precise control over the metallic dimensions, including shell thickness (Linic et al.), gap size (Halas et al.; Cortes et al.; Jaw-Min Nam et al.), large area (Cortes metasurface and supercrystals), and coupling strength (Misawa et al.), achieved through techniques such as lithography and colloidal self-assembly. Among these studies, the correlation between optical enhancements, reaction efficiency, and surface chemistry has been better understood. The authors claim there may be room that the optical enhancement be higher than previous study (which I highly doubt in a rigorous manner of fair comparison), but at least the impact on the kinetic characterization of the catalytic performance is lacking in this article as the support.

> While previous studies indeed explored antenna-reactors geometries (as we cite), our study introduces a unique atomic-monolayer coating of the plasmonic nanocavity that offers unprecedented control over the metallic gap (1-2nm) and the highest catalytic atom efficiency. This control allows us to achieve theoretical limits of plasmon-mediated chemical reactions, as demonstrated by our experimental results. Specifically, we have shown significant enhancements in catalytic activity for ethanol oxidation, CO₂ reduction, and hydrogen evolution reactions. These results are supported by rigorous quantitative measurements, including catalytic current analysis, which highlight the superior performance of our system. We acknowledge that a more detailed kinetic characterization, such as rate constants or activation energies, would provide additional insights. However, the photocurrent data we have presented serves as a robust and quantitative measure of the reaction rate, which is a key aspect of kinetic characterization. We believe that this data, along with the other results in the manuscript, provides sufficient evidence to support our claims of enhanced catalytic performance, but following this comment, now better clarify how comparisons might be best achieved. We also emphasize the surprising result here that it is possible to create plasmonic structures that behave optically as Au, but chemically as Pd – this is completely different from previous approaches in which the Pd nanoparticles significantly worsen the Au plasmonic response.

2. The electrochemical cyclic voltammetry (CV) and synchrotron X-ray photoelectron spectroscopy (XPS) are valuable bulk analysis techniques but do not provide spatial resolution down to sub-atomic layers. In contrast, high-resolution transmission electron microscopy (HR-TEM) offers a more direct and feasible approach for achieving the high spatial resolution required, providing crucial evidence to support the claim. Although HR-TEM imaging can be challenging, it remains a reasonable and effective technique for this purpose. Additionally, low-dose imaging and focused ion beam (FIB) milling are well-established techniques that can significantly aid in imaging such metallic structures.

> We understand the referee's preference for HR-TEM for direct structural characterization. However, due to the extremely narrow nanogap (1-2 nm) in our system and the unique geometry involving an 80nm Au nanoparticle and a bulk substrate, HR-TEM is unfeasible (we now tried with leading

collaborators). The metallic gap forms directly between the nanoparticle and bulk substrate, making it extremely challenging to resolve monolayer atoms inside with atomic resolution. Additionally, the high-energy electron beam required for optimal resolution in HR-TEM is suspected to cause irreversible damage to the delicate monolayer structures, raising concerns about the reliability of the data obtained.

Beyond this, HR-TEM provides only a sectional view of the structural phases, which limits our ability to capture the overall lateral surface characteristics, especially considering the statistical variations across a large number of NPoM geometries. Techniques like low-dose imaging and FIB milling, while useful in some contexts, are not well-suited for our specific system due to the significant restructuring of gold (which is rather ductile for these techniques), their complexity and the nanogap scale. Instead, we have employed CV and XPS as complementary techniques to provide indirect but reliable evidence of the nanostructure. These methods are well-suited for our system and have been validated by extensive literature. While we acknowledge the limitations of these techniques, we believe they provide sufficient evidence to support our claims, especially given the challenges associated with direct imaging of such small structures.

In response to this suggestion, we have thus clarified and rephrased the relevant sections to better address these points and ensure the robustness and clarity of our findings.

3. Table 1 is more of a general experience rather than a well-designed guide for the application. Engineered atomic transition metal layers, depending on their local *d*-band structure, can effectively shift the chemical affinity for different reactions. This can cause the surface to switch from being reactive to behaving as a poison, as described by the Sabatier principle.

> Indeed we appreciate the referee's comment on Table 1. The table is intended to illustrate the alchemical space for atomic glazing on Au substrates, specifically addressing how easily different metals can be glazed onto NPoMs. Since in Fig.6 taken from Table 1, the *x*-axis is the relative binding energy between the glazing metal and Au substrate, while the *y*-axis is the lattice mismatch, this quantitatively provides key information which determines the stability and feasibility of forming high-quality monolayer glazed films. This framework is based on both experimental findings and theoretical considerations, providing a practical roadmap for selecting metals that can be effectively glazed onto Au substrates.

Regarding the referee's comment on the local *d*-band structure and its influence on reactivity, we agree that atomic layer coating can shift the *d*-band center (also affecting the chemical affinity), which often enhances the reactivity of the surface (depending on the lattice mismatch). However while the referee suggests that this could cause the surface to switch from being reactive to behaving as a poison, we note that this is not typically the case for the systems we have studied. In fact, many studies (*Angew. Chem. Int. Ed.* 2005, 44, 2080; *Science* 2007, 315, 220; *Angew. Chem.* 2005, 117, 2170; *J. Am. Chem. Soc.* 2009, 131, 17298), including our own, demonstrate that atomic layer coatings enhance catalytic activity rather than poisoning the surface. For example, the glazing of Pd, Pt, or Cu onto Au substrates has been shown to significantly improve catalytic performance in reactions such as ethanol oxidation, CO₂ reduction, and hydrogen evolution. We thus add a sentence to discuss this point.

Reviewer 5

1. Cyclic Voltammetry and Monolayer Calibration.

I agree with the referees that connecting cyclic voltammetry data to discrete, stepwise nanostructural changes from sub-monolayers to multilayers remains challenging. Without error bars, it is difficult to ascertain whether these surfaces are uniform or exhibit variations at the level of individual NPoMs. Additionally, the relationship between monolayer coverage and the observed signals may vary between experiments, raising questions about how this approach can be calibrated into a reliable "monolayer gauge." While the authors make a strong case for monolayer control using self-terminating underpotential deposition (supported by relevant literature), a more detailed discussion

of this methodology's limitations and its reproducibility would be beneficial. The SERS spectra effectively confirm the presence of Pd (mono)layers, but the data could be further contextualized to enhance clarity.

> We thank the referee for their thoughtful comments. The CV data presented in our study provides strong evidence for the existence of monolayer foreign atoms on the surface (where the reactions occur). The continuous and systematic evolution of the CV curves with increasing coverage (from sub-monolayer to multilayer) is a hallmark of monolayer formation, as discontinuous or irregular changes would indicate non-uniform or mixed structures. This behavior is consistent with the well-established underpotential deposition (UPD) process, which ensures precise monolayer control.

Unlike methods such as HR-TEM, which can only provide information about a very few or single structure and are challenging to apply to systems with nanogaps, CV offers a bulk-sensitive approach that reflects the average behavior of the entire surface. This is particularly important for ensuring uniformity across a large number of NPoM structures. While defects are inherently unavoidable on any surface, they should not alter our conclusions.

While we acknowledge that CV data alone cannot resolve atomic-scale variations, the combination of CV with SERS spectra (which confirm the presence of Pd monolayers) provides a robust and complementary characterization of the system. As suggested, we have now also included a more detailed discussion of this part in the revised manuscript, emphasizing how experimental conditions are carefully controlled to ensure reliable monolayer formation.

2. Clarity on Experimental Scale and Scope

It would be helpful if the authors clarified how many NPoMs are involved in each of the different measurements (electrochemistry, SERS, XPS) - sometimes I found this to be not very clear.

> The referee points out an important omission. The SERS of phenyl isocyanide molecules is performed using Au nanoparticle aggregates - a laser spot of diameter of $1\mu\text{m}$ covers >100 nanoparticles. All the other SERS measurements are done on single NPoMs with a statistical analysis and numbers specified. In the electrochemical measurements, the nanoparticles are deposited on an electrode of 5mm in diameter that spans millions of nanoparticles. The XPS measurements are done on an atomically-flat Au film with Pd glazed on top. We thus revised the manuscript to explicitly state the scale of each measurement, ensuring clarity for readers.

3. NPoM Density and Catalytic Activity

Does the density of NPoMs influence photocatalytic activity? How was the working density of NPoMs controlled in these experiments? Has the spatial distribution of particles been characterized to ensure uniformity and to rule out potential aggregation effects?

> These are good points to clarify. The density of NPoMs can indeed influence photocatalytic activity, as it affects both light absorption and reactant diffusion. In our experiments, we carefully optimized the NPoM density to maximize catalytic performance while minimizing aggregation. The density of NPoMs was controlled by adjusting the concentration of nanoparticles during the drop-casting process. This allowed us to achieve a uniform distribution of NPoMs across the substrate, ensuring consistent experimental conditions. To confirm uniformity and rule out aggregation effects, we have now included a new SEM image (as Supplementary Figure 19) that characterizes the spatial distribution of NPoMs. These images demonstrate a well-distributed and non-aggregated arrangement of nanoparticles, supporting the reliability of our results.

Supplementary Figure 19. SEM image showing density of Au NPoMs used in photocatalytic measurements.

4. Structural Characterization and Facet Size

The authors infer significant increases in facet size from indirect measurements, which is an important observation. However, direct imaging of these nanoparticle formations—possibly after dislodging them from the substrate—could provide more definitive evidence. Have the authors considered pursuing this approach, or is it technically prohibitive?

> The formation of facet during the fabrication of NPoMs has been directly imaged in prior literature (see **Fig.R1** below) using HR-TEM. However, the direct imaging of facet growth remains extremely difficult for the techniques currently available. Firstly, the nanoparticles are not perfect spheres and already have random facets initially, making it difficult to identify and probe specific facets. Secondly, dislodging the nanoparticles from the substrate without altering their structure is extremely difficult, as the process introduce artifacts and damages delicate facets. Moreover, tracking the same particle through deposition, optical measurements, and re-imaging is not feasible due to the random distribution of particles during drop-casting. Given these challenges, we rely on optical measurements (plasmonic scattering spectroscopy), which provide indirect but highly consistent and reproducible evidence of facet size changes.

Figure R1. HR-TEM image of a NPoM cross-section. [from *Light: Science & Applications* (2018) 7:56]

Chen, W., Zhang, S., Kang, M. et al. Probing the limits of plasmonic enhancement using a two-dimensional atomic crystal probe. *Light Sci Appl* 7, 56 (2018). <https://doi.org/10.1038/s41377-018-0056-3> (CC BY 4.0)

5. Single-Molecule Turnover and Catalysis

Their SERS data suggests significant potential for monitoring single NPoM systems. In the future, the authors could explore the possibility of tracking catalytic activity at the level of single NPoMs, perhaps even down to single-molecule turnover. This could require integrating controlled picocavities into the current system. A brief discussion of this direction could enrich the manuscript's outlook.

> We thank the referee for this ambitious suggestion. We agree that exploring catalytic activity at the single-molecule level is an exciting and highly promising direction. Controlling picocavities within the NPoM system presents intriguing opportunities but also significant technical challenges. Consistently triggering and maintaining stable picocavities of sub-nm scale size remains unfeasible, as the

underlying mechanisms of picocavity formation are not yet fully understood (*Nano Lett.* 2022, 22, 5859). Nevertheless, our current data, which demonstrate the glazed NPoMs have high catalytic and optical activity, lays a strong foundation for future investigations at the single-molecule scale. In the revised manuscript, we include a brief discussion of this potential future direction, as it offers a compelling and innovative avenue for advancing the field of plasmonic catalysis.